# Periodic Mesoporous Organosilica Nanoparticles with BOC Group, towards HIFU Responsive Agents

**DOI:** 10.3390/molecules25040974

**Published:** 2020-02-21

**Authors:** Hao Li, Carolina Gascó, Anthony Delalande, Clarence Charnay, Laurence Raehm, Patrick Midoux, Chantal Pichon, Roser Pleixats, Jean-Olivier Durand

**Affiliations:** 1ICGM, Univ Montpellier, CNRS, case 1701, Place Eugène Bataillon, CEDEX 05, 34095 Montpellier, France; Hao.Li@uab.cat (H.L.); clarence.charnay@umontpellier.fr (C.C.); laurence.raehm@umontpellier.fr (L.R.); 2Departament de Química and Centro de Innovación en Química Avanzada (ORFEO-CINQA), Facultat de Ciències, Universitat Autònoma de Barcelona, UAB Campus, C/dels Til.lers, 08193 Cerdanyola del Vallès (Barcelona), Spain; Carolina.Gasco@uab.cat; 3Center for Molecular Biophysics, (CBM, UPR 4301), Rue Charles Sadron, 45071 Orléans, France; anthony.delalande@cnrs-orleans.fr (A.D.); patrick.midoux@cnrs-orleans.fr (P.M.); chantal.pichon@cnrs-orleans.fr (C.P.)

**Keywords:** HIFU, periodic mesoporous organosilica nanoparticles, cancer, CO_2_

## Abstract

Periodic Mesoporous Organosilica Nanoparticles (PMONPs) are nanoparticles of high interest for nanomedicine applications. These nanoparticles are not composed of silica (SiO_2_). They belong to hybrid organic–inorganic systems. We considered using these nanoparticles for CO_2_ release as a contrast agent for High Intensity Focused Ultrasounds (HIFU). Three molecules (**P1**–**P3**) possessing two to four triethoxysilyl groups were synthesized through click chemistry. These molecules possess a tert-butoxycarbonyl (BOC) group whose cleavage in water at 90–100 °C releases CO_2_. Bis(triethoxysilyl)ethylene **E** was mixed with the molecules **Pn** (or not for **P3**) at a proportion of 90/10 to 75/25, and the polymerization triggered by the sol-gel procedure led to PMONPs. PMONPs were characterized by different techniques, and nanorods of 200–300 nm were obtained. These nanorods were porous at a proportion of 90/10, but non-porous at 75/25. Alternatively, molecules **P3** alone led to mesoporous nanoparticles of 100 nm diameter. The BOC group was stable, but it was cleaved at pH 1 in boiling water. Molecules possessing a BOC group were successfully used for the preparation of nanoparticles for CO_2_ release. The BOC group was stable and we did not observe release of CO_2_ under HIFU at lysosomal pH of 5.5. The pH needed to be adjusted to 1 in boiling water to cleave the BOC group. Nevertheless, the concept is interesting for HIFU theranostic agents.

## 1. Introduction

The use of Periodic Mesoporous Organosilica Nanoparticles (PMONPs) has grown a lot in the last decade, particularly for biological applications, and the field has been recently reviewed [1,2,3]. Hollow Mesoporous Organosilica Nanoparticles have been more recently described and reviewed as well [4]. PMONPs are synthesized from trialkoxypolysilylated organic precursors and usually silica sources are not used for their preparation. PMONPs possess improved properties compared to Mesoporous Silica Nanoparticles (MSN), which have been considerably reported for their nanomedicine applications [5,6,7,8,9,10,11,12]. Indeed, PMONPs can be loaded with hydrophobic or hydrophilic molecules and present better hemolytic properties than MSN [13,14]. PMONPs are more stable than MSN in medical relevant conditions [15], but, with disulfide bridges in their structure, their degradability can be triggered by glutathione [16].

Functionalized MSN [17,18] and PMONPs [19,20,21,22] were combined with high-intensity focused ultrasound (HIFU) for theranostic applications of cancer. Indeed, HIFU has been used for the ablation of pathological lesions and tumors [23,24]. Bubbles are usually the most developed contrast agent for HIFU [25,26,27]. Furthermore, recent efforts have been dedicated to the preparation of carriers able to generate CO_2_ bubbles [28]. In the course of our studies on PMONPs, we were interested in nanomaterials possessing the tert-butoxycarbonyl (BOC) group. Indeed, BOC, as an amine protective group, can be quickly removed in water at 90–100 °C, generating CO_2_ [29,30]. As HIFU can heat up to 70–80 °C [31,32], we thought of synthesizing PMONPs from polytrialkoxysilylated BOC-functionalized molecules in order to develop PMONP-based contrast agents for HIFU.

We present here our studies of the syntheses of three bis(triethoxysilylated) **P1** and tetra(triethoxysilylated) molecules **P2**, **P3** (Scheme 1) possessing a BOC group, and their reaction through the sol-gel procedure with bis(triethoxysilyl)ethylene **E**. The materials called **E-Pn 90/10 PMONPs** or **E-Pn 75/25 PMONPs** correspond to a sol-gel reaction with 90% molecule **E** and 10% molecule **Pn,** or 75% molecule **E** and 25% molecule **Pn,** respectively. **E PMONPs** was also synthesized and corresponds to 100% of molecule **E**. The materials were analyzed with different techniques and showed mesoporosity at a 10% proportion of **P1**–**P3** molecules. When the proportion increased, no more porosity was observed. Pure **P3 PMONPs** with the tetra(triethoxysilylated) molecule **P3** are also described and showed mesopores of 5.6 nm. PMONPs are, therefore, promising materials for theranostic applications.

## 2. Results and Discussion

We envisaged to synthesize the bis(triethoxysilylated) and tetra(triethoxysilylated) molecules **P1**–**P3** possessing a BOC group through a copper-catalyzed azide-alkyne cycloaddition reaction (CuAAC) under anhydrous conditions [33], by treatment of the appropriate dialkyne with 3-azidopropyltriethoxysilane, 3-azido-*N*,*N*-bis(triethoxysilylpropyl)propanamine or 2-azido-*N*,*N*-bis(triethoxysilylpropyl)ethanamine [34], respectively (Scheme 1).

*N*-Boc protection of propargyl amine **1** furnished carbamate **2** in 91% yield, which was reacted with propargyl bromide in the presence of NaH in anhydrous THF to afford dialkyne **3** in quantitative yield (Scheme 2) [35].

The bis-silylated precursor **P1** was obtained by the reaction of dialkyne **3** with (3-azidopropyl)triethoxysilane under copper (I) catalysis under anhydrous conditions. First, the copper catalyst [CuBr(PPh_3_)_3_] was used, but we found that it was difficult to remove it from the crude mixture by washing with different solvents after the reaction. Then, CuI was chosen as a copper source and tris[(1-benzyl-1*H*-1,2,3-triazol-4-yl)methyl]amine (TBTA) as a ligand that stabilizes Cu(I) ions in a 1:1 mixture of anhydrous Et_3_N/anhydrous THF at 50 °C overnight (Scheme 3). After the work-up and purification, the bis(silylated) precursor **P1** was obtained at 73% yield. The purification was performed by washing with hot anhydrous pentane several times. As the product remains partially dissolved in pentane, the isolated yield is lower.

Similarly, the other two silylated precursors **P2** and **P3** were obtained, both at 96% yield, by reacting the corresponding bis(silylated azide) with dialkyne **3** under the same reaction conditions (Scheme 4).

With those precursors in hands, we next performed the sol-gel procedure in highly diluted conditions with 1,2-bis(triethoxysilyl)ethene (**E**) as the major reagent. First, the synthesis was performed in Mili-Q water with cetyltrimethylammonium bromide (CTAB) micellar template and sodium hydroxide catalysis at 80 °C. The template was prepared at this temperature for 50 min with a stirring speed adjusted at 1400 rpm. Then, 1,2-bis(triethoxysilyl)ethene (in a proportion of 90% or 75%) was added with **P1**–**P3** (in a proportion of 10% or 25%) and the mixture reacted for two hours. Nanoparticles were collected by centrifugation and the template was removed by washing with an ethanolic solution of ammonium nitrate; the resulting material was washed successively with ethanol, Mili-Q water and ethanol. Herein, two different ratios of **E**-**Pn** were tested—90/10, 75/25—in order to study their influence on the NPs’ size and morphology.

All the mixed periodic mesoporous organosilica nanoparticles were characterized by TEM, nitrogen-sorption measurements (BET), *p*-XRD, FTIR, ^13^C-CP MAS solid-state NMR, DLS, zeta-potential, TGA and DSC. Some physical data are given in Table 1.

The morphologies of **E-Pn** were first analyzed by TEM (Figure 1), and for the majority of the materials rodlike-shape nanoparticles were obtained ranging from 200 nm for (**E-P1**, **E-P3 90/10**) to 300 nm long for **E-P2**. For **E-P3 75/25**, nanoparticles of 150 nm were observed. The hydrodynamic diameters of the nanorods were investigated by DLS, (Table 1) and the obtained diameters were in agreement with the sizes determined by TEM. **E-P2** showed higher hydrodynamic diameters than **E-P1** and **E-P3**. The hydrodynamic diameter of **E-P3 75/25** was lower than (**E-P1**, **E-P2**, **E-P3 90/10**) in agreement with a smaller size and particle morphology compared to nanorods. The zeta potentials were determined in water (Table 1). **E-P1** showed negative zeta potential, in agreement with the presence of Si-O^−^ at the surface of the nanorods. The zeta potential of **E-P2** and **E-P3** was positive because the amino groups of **P2** and **P3** were protonated below pH 7.

The N_2_ adsorption-desorption isotherms were next examined (Figure 2). All the materials showed an isotherm between type I and type IV, and between microporous and mesoporous systems. Nanorods from **E** (already synthesized) [36] showed a very high specific surface area of 800 m^2^·g^−1^ with an average small pore size of 2.5 nm. When incorporating **P1**, **E-P1 90/10** showed a decrease in specific surface area to 683 m^2^·g^−1^ and a small increase in pore size to 2.8 nm. The incorporation of **P2** and **P3** at a proportion of 90/10 led to a decrease in specific surface area to 500 m^2^·g^−1^ with an increase of pore size to 2.7 and 2.8 nm, respectively. Note that when increasing the proportion of precursors **P1-3** to 75/25, materials **E-P1-3**, showed no adsorption of N_2_ any more at 78 K, which suggests that the porosity is not accessible to N_2_ at this temperature.

Small Angle X-rays diffraction of nanorods from **E** [36] (Figure 3) showed the d(1,0,0) Bragg peak at 2.2 θ, with the satellites at d(1,1,0) 4.0 θ and d(2,0,0) 4.6 θ, characteristics of a well-ordered hexagonal mesoporosity. The other materials are more disordered with a wormlike mesoporosity and a broad non-structured band (see XRD of **E-P1 90/10** and **E-P1 75/25** as representative examples).

FTIR were recorded for all the precursors and materials. A representative example is given in Figure 4 with **P2**, **E-P2 90/10** and **E-P2 75/25**. **P2** showed the presence of the BOC group at 2973 cm^−1^ (νCH_3_), the carbon chains at 2927 and 2883 cm^−1^ (νCH_2_), the carbonyl of the carbamate at 1696 cm^−1^ (νC=O), the triazole νCH at 1457–1386 cm^−1^, the silicon–oxygen bonds at 1073 cm^−1^ (νSi-O). All the bands were essentially the same in the materials, with a shift in the carbonyl to 1675–1672 cm^−1^ showing a different environment of the BOC group in the material. A broad band between 1200–1000 cm^−1^ ((νSi-O-Si), characteristic of a well-condensed material, was observed.

^13^C-CPMAS solid-state, NMR were also recorded. Spectrum (**E-P2 75/25**) is representative (Figure 5) and showed the characteristic signals (in ppm) at 155.9 (C=O), 145.3 (C=C, Cq triazole), 124.0 (CH triazole), 80.4 (Cq, *t*Bu), 55.2 (CH_2_-N, CH_2_-triazole), 28.6 (CH_3_*t*Bu) 21.6(-CH_2_-), 11.2 (CH_2_-Si).

TGA analyses were performed (Figure 6). The first loss of mass corresponded to water (up to 120 °C). From 150 to 220 °C, the loss of the BOC group was observed. The BOC group was very stable in the materials in dry state, as no loss of mass was observed before 120 °C. Then, the decomposition of the organic part of the materials was shown.

We then performed the synthesis of **P3 PMONPs**. For this, **P3** was used as the only precursor (no silica source). CTAB was dissolved in aqueous NaOH at 80 °C in highly diluted conditions for 50 min, then the precursor **P3,** dissolved in EtOH, was rapidly added. The condensation mixture was stirred at 80 °C for 24 h, then extraction of CTAB and drying of the sample was performed as above. The nanoparticles were analyzed using different techniques.

First, TEM images (Figure 7) showed spherical monodisperse nanoparticles (100 nm diameter) which seem to be composed of sheets of organosilica. The nanoparticles are clearly porous on TEM images. The porosity was confirmed with N_2_ sorption experiments (BET). Indeed, a type IV isotherm was obtained (Figure 8) and the porosity was determined to be 5.6 nm. The hydrodynamic diameter was observed at 269 nm and zeta potential was positive, in agreement with the protonation of tertiary nitrogen atoms (Table 1). The precursor **P3** was not damaged after reaction, as shown by FTIR spectroscopy (Figure 9) with the carbonyl group observed at 1682 cm^−1^.

**E-P3 75/25 PMONPs** were selected for the release of CO_2_, first under water conditions at 100 °C. A total of 50 mg of **E-P3 75/25 PMONPs** were added to 10 mL of distilled water, then the mixture was quickly heated to 100 °C and stirred at this temperature for 30 min. The experiment was performed at two different pH: 7 and 1. At pH 7, we did not observe any bubbles; at pH 1 we could clearly see a lot of bubbles in the mixture (Figure 10). Then, **E-P3 75/25 PMONPs** were collected by centrifugation and washed with EtOH. Finally, **E-P3 75/25 PMONPs** were dried for FTIR analyses. At pH 7, the BOC group was still present, but at pH 1 the absorption of *t*Bu (2987 cm^−1^) and C=O (1680 cm^−1^) disappeared in the spectrum (Figure 11). These results confirmed the successful release of CO_2_ in pH 1 water at 100 °C.

We then tried to carry out BOC removal under HIFU conditions. The different nanoparticles were suspended in water at a concentration of 10 mg/mL and submitted to HIFU, but due to the stability of BOC group in the materials, we did not observe any cleavage of the BOC group (data not shown).

## 3. Materials and Methods

### 3.1. General Remarks

The ^1^H- and ^13^C-NMR spectra in solution were recorded on Bruker DPX-360 MHz or Avance-II 600 MHz (Bruker Biospin, Rheinstetten, Germany) and are referenced to solvent signals (CDCl_3_: δ = 7.26 ppm). All the spectra were calibrated using the residual solvent signal (CHCl_3_, δ_H_, 7.26 and δ_C_, 77.16 ppm). Chemical shift data are expressed in ppm and coupling constant (*J*) values in Hz. The multiplicity of peaks is abbreviated as s (singlet), d (doublet), t (triplet), q (quartet) and dd (doublet of doublets). These NMR instruments belong to the *Servei de Ressonància Magnètica Nuclear* of the *Universitat Autònoma de Barcelona* (UAB). The ^13^C-CP-MAS solid state NMR spectra were obtained from a Varian VNMRS 300 MHz instrument (Les Ulis, France) which belongs to the *University of Montpellier*; the repetition time was 5 s with contact times of five milliseconds. From the *Servei d’Anàlisi Química* of the UAB, the following experimental data were acquired: infra-red spectra (IR), mass-spectrometry (MS) and elemental analysis. IR spectroscopy was recorded with a Bruker Tensor 27 spectrometer (Bruker Biosciences Española, Madrid, Spain) using a Golden Gate ATR module with a diamond window. Low- and high-resolution mass spectra were obtained by direct injection of the sample with electrospray techniques in a Hewlett-Packard 5989A and *microTOF-Q* instruments (Bruker Biosciences Española, Madrid, Spain), respectively. Elemental analysis of C, N and H were performed using Flash 2000 Organic Elemental analyser (Barcelona, Spain) of Thermo Fisher Scientific with BBOT as an internal standard. Transmission Electron Microscopy (TEM) was performed on JEM-2011 Electron Microscope 200 Kv (JEOL Ltd., Akishima, Tokyo, Japan), which belongs to the *Servei de Microscòpia* of UAB. Powder X-ray diffraction (*P*-XRD) was performed in the *Servei de Difracció de Raigs X* of UAB with X’Pert Power from PANalytical (Almelo, The Netherlands) 45 Kv/40 mA, K_α_ 1.5419 Å with a cooper anode. Dynamic light scattering (DLS) and Zeta potential were obtained from *University of Montpellier*. DLS analyses were performed using a Cordouan Technologies DL 135 (Pessac, France) particle size analyzer instrument at *University of Montpellier*. Zeta potential measurements were performed on MALVERN Instruments ZETA SIZER Nano series (Orsay, France) with 1–3 mg samples dispersed in 5 mL of aqueous solution (distilled water or 0.01 M NaCl solution) between pH 4 and 9. The surface areas were determined by the Brunauer-Emmet-Teller (BET) method from N_2_ adsorption–desorption isotherms obtained with a Micromeritics ASAP2020 analyzer (Mérignac, France) which belongs to the *University of Montpellier* after degassing samples for 12 h at 80 °C under vacuum. The total pore volumes were evaluated by converting the volume adsorbed at p/p^o^ 0.98 to the volume of liquid adsorbed (single point adsorption total pore volume of pores less than 4000 Å at p/p° ≈ 0.98). The pore size distributions for PMONPs were determined from the desorption branch using the Barrett–Joyner–Halenda (BJH) method, which relies on the Kelvin equation to relate the width of the pores to the condensation pressure. When required, experiments were carried out with standard high vacuum and Schlenk techniques. Chromatographic purifications were performed under N_2_ pressure using 230–400 mesh silica gel (flash chromatography). Dry solvents and reagents were obtained following standard procedures: triethylamine was distilled over CaH_2_; DMF, CH_3_CN and pyridine were dried by molecular sieves, THF, CH_2_Cl_2_ and pentane were from solvent processing equipment (*PureSolv*, Innovative Technology, Newburyport, MA, USA). Cetyltrimethylammonium bromide (CTAB), sodium hydroxide, ammonium nitrate (NH_4_NO_3_), and potassium bromide were purchased from Sigma-Aldrich. Absolute ethanol was purchased from Fisher Chemicals and hydrochloric acid from VWR PROLABO.

### 3.2. Synthesis of **P1**

To a dry 100 mL Schlenk flask equipped with a stir bar and under Argon atmosphere, CuI (7.60 mg, 0.04 mmol), TBTA (21.20 mg, 0.04 mmol) and anhydrous THF (20 mL) were added. The resulting mixture was stirred for 30 min, then (3-azidopropyl)triethoxysilane (988.5 mg, 4.0 mmol), *tert*-butyl di(prop-2-yn-1-yl)carbamate (463.46 mg, 2.4 mmol) and anhydrous Et_3_N (1.0 mL, 0.73 g/mL 7.2 mmol) were added by using a syringe. The resulting mixture was stirred at 50 °C (Argon atmosphere) until *tert*-butyl di(prop-2-yn-1-yl)carbamate was fully consumed (16 h, reaction monitored by TLC). Then, the solvent was evaporated under reduced pressure and the residue was washed with cold anhydrous pentane to remove some remaining azide. Hot anhydrous pentane was added to the insoluble fraction to extract the disilylated product; this digestion with hot pentane was repeated several times and the pentane extracts were concentrated under reduced pressure to provide the pure product as a colorless oil (999 mg, 73% yield). ^1^H-NMR (360 MHz, CDCl_3_) δ 7.53 (s, 1H), 7.42 (s, 1H), 4.51 (s, 4H), 4.29 (t, *J* = 7.2 Hz, 4H), 3.78 (q, *J* = 7.2 Hz, 12H), 2.00–1.96 (m, 4H), 1.42 (s, 9H), 1.17 (t, *J* = 7.2 Hz, 18H), 0.56 (t, *J* = 7.2 Hz, 4H); ^13^C-NMR (91 MHz, CDCl_3_) δ 155.2, 122.7, 121.9, 80.3, 58.5, 58.4, 52.4, 28.4, 24.2, 18.2, 7.4. IR (film) 3132.4, 2973.6, 2927.2, 1692.6, 1390.2, 1163.5, 1073.1, 954.5, 783.2 cm^−1^. MS (ESI) *m*/*z*: 688.4, 632.3, 588.3, 558.3; HRMS (ESI) *m*/*z* [M + Na]^+^ calcd. for C_29_H_57_N_7_O_8_Si_2_Na: 688.3880, found: 688.3877.

### 3.3. Synthesis of **P2**

To a dry, 100 mL Schlenk flask equipped with a stir bar and under Argon atmosphere, CuI (7.60 mg, 0.04 mmol), TBTA (21.20 mg, 0.04 mmol) and anhydrous THF (20 mL) were added. The resulting mixture was stirred for 30 min, and then *N*-(2-azidoethyl)-3-(triethoxysilyl)-*N*-(3-(triethoxysilyl)propyl)propan-1-amine (1.98 g, 4.0 mmol), *tert*-butyl di(prop-2-yn-1-yl)carbamate (463.46 mg, 2.4 mmol) and anhydrous Et_3_N (1.0 mL, 0.73 g/mL, 7.2 mmol) were added by using a syringe. The resulting mixture was stirred at 50 °C (Argon atmosphere) until *tert*-butyl di(prop-2-yn-1-yl)carbamate was fully consumed (20 h, reaction monitored by TLC). Then, the solvent was partially evaporated under reduced pressure, the remaining solution was filtered through Whatman^®^ membrane filters PTFE (pore size 0.2 μm, diam. 47 mm) and the filtrate evaporated at reduced pressure to afford the final product as colorless oil (2.27 g, 96% yield). ^1^H-NMR (360 MHz, CDCl_3_) δ 7.61 (s, 1H), 7.57 (s, 1H), 4.51 (s, 4H), 4.36 (t, *J* = 7.2 Hz, 4H), 3.80 (q, *J* = 7.2 Hz, 24H), 2.88 (t, *J* = 7.2 Hz, 4H), 2.47 (t, *J* = 7.2 Hz, 8H), 1.53–1.45 (m, 17H), 1.20 (t, *J* = 7.2 Hz, 36H), 0.54 (t, *J* = 7.2 Hz, 8H); ^13^C-NMR (91 MHz, CDCl_3_) *δ* 155.1, 128.9, 127.9, 80.1, 58.2, 57.0, 54.1, 48.8, 28.3, 28.1, 20.2, 18.1, 7.7. IR (film): 2972.4, 2927.2, 2882.9, 1695.7, 1457.0, 1389.6, 1164.1, 1073.0, 952.5, 770.1 cm^−1^. MS (ESI) *m*/*z*: 1204.7, 1182.7, 688.4, 591.9, 413.3; HRMS (ESI) *m*/*z* [M + Na]^+^ calcd. for C_51_H_107_N_9_O_14_Si_4_Na: 1204.6906, found: 1204.6894.

### 3.4. Synthesis of **P3**

To a dry, 100 mL Schlenk flask equipped with a stir bar and under Argon atmosphere, CuI (7.60 mg, 0.04 mmol), TBTA (21.20 mg, 0.04 mmol) and anhydrous THF (20 mL) were added. The resulting mixture was stirred for 30 min, then 3-azido-*N*,*N*-bis(3-(triethoxysilyl)propyl)propan-1-amine (2.03 g, 4.0 mmol), *tert*-butyl di(prop-2-yn-1-yl)carbamate (463.46 mg, 2.4 mmol) and anhydrous Et_3_N (1.0 mL, 0.73 g/mL, 7.2 mmol) were added by using a syringe. The resulting mixture was stirred at 50 °C (Argon atmosphere) until *tert*-butyl di(prop-2-yn-1-yl)carbamate was fully consumed (20 h, reaction monitored by TLC). Then, the solvent was partially evaporated under reduced pressure, the remaining solution was filtered through Whatman^®^ membrane filters PTFE (pore size 0.2 μm, diam. 47 mm) and the filtrate was evaporated at reduced pressure to afford the final product as colorless oil (2.33 g, 96% yield). ^1^H-NMR (360 MHz, CDCl_3_) δ 7.55 (s, 1H), 7.47 (s, 1H), 4.53 (s, 4H), 4.36 (t, *J* = 7.2 Hz, 4H), 3.81 (q, *J* = 7.2 Hz, 24H), 2.48–2.36 (m, 12H), 2.01 (t, *J* = 7.2 Hz, 4H), 1.52–1.46 (m, 17H), 1.21 (t, *J* = 7.2 Hz, 36H), 0.57 (t, *J* = 7.2 Hz, 8H); ^13^C-NMR (91 MHz, CDCl_3_) *δ* 155.2, 129.1, 128.0, 80.3, 58.3, 57.1, 56.8, 50.9, 48.5, 28.4, 26.8, 20.1, 18.3, 7.9. IR (film) 2972.7, 2926.8, 2883.2, 1695.4, 1455.6, 1389.7, 1164.0, 1072.8, 952.5, 770.3 cm^−1^. MS (ESI) *m*/*z*: 1232.7, 1210.7, 816.5, 702.4, 605.9, 509.3, 438.3; HRMS (ESI) *m*/*z* [M + Na]^+^ calcd. for C_53_H_111_N_9_O_14_Si_4_Na: 1232.7220, found: 1232.7195.

### 3.5. Synthesis of **E-BTSE**

Grubbs Catalyst^™^ 1st Generation C823 (PCy_3_)_2_Cl_2_Ru=CHPh (82.30 mg, 0.1 mmol) and vinyltriethoxysilane (VTES) (19.02 g, 100 mmol) were added to a Schlenk flask under argon. After the solution was stirred for one hour and the mixture was refluxed for 24 h, unreacted VTES was distilled off. Subsequently, BTSE was vacuum distilled to give a clear colorless liquid (12.53 g, 71% yield, 4 mbar, 124 °C). *E*-BTSE was identified by ^1^H-NMR as a diastereoisomerically pure product (100% *E*). ^1^H-NMR (360 MHz, CDCl_3_) *δ* 6.66 (s, 2 H; *E*-isomer), 3.83 (q, *J* = 7.2 Hz, 12 H), 1.23 (t, *J* = 7.2 Hz, 18 H).

### 3.6. Preparation of **E-BTSE PMONPs**

In an open 250 mL round-bottom flask without reflux condenser, a solution of cetyltrimethylammonium bromide (CTAB, 250 mg, 0.686 mmol) was placed in Mili-Q water (120 mL) and then 875 µL of 2 M NaOH was added (1.75 mmol of NaOH). The mixture was stirred at 1000 rpm at 80 °C for 50 min. Then, the stirring speed was enhanced to 1400 rpm and 1,2-bis(triethoxysilyl)ethene (BTSE, 800 μL, 2.29 mmol) was added rapidly under stirring. The condensation process was conducted for two hours at 80 °C. The suspension was cooled to room temperature while stirring and the NPs were collected by centrifugation (13,500 rpm in 6 × 25 mL flasks tubes for 45 min). The supernatant was removed and an aliquot of the solid was taken and dried for FTIR. In order to remove the surfactant, 10 mL of an alcoholic solution of ammonium nitrate [NH_4_NO_3_, 6 g/L in 96% EtOH] was added to each tube. The six tubes were sonicated for 30 min at 50 °C, then cooled and centrifuged (30 min at 13,500 rpm at 25 °C), the supernatant was discarded. This NH_4_NO_3_ washing was performed three times. Each solid in the tubes was washed successively with 96% ethanol, Mili-Q water, 96% ethanol using the same protocol (30 min at 50 °C sonication, centrifugation). The final product was dried for few hours under vacuum at room temperature. The **E PMONPs** were obtained as a white solid. At the same time, another aliquot of the solid was taken and dried for FTIR. It was compared to the IR of the sample before removal of the surfactant and after removal to check whether the CH_2_ bands of CTAB had completely disappeared.

### 3.7. Preparation of **E-Pn 90/10 PMONPs**

In an open 250 mL round-bottom flask without reflux condenser, a solution of CTAB (250 mg, 0.686 mmol) was placed in Mili-Q water (120 mL) and then 875 µL of 2 M NaOH was added (1.75 mmol of NaOH). The mixture was stirred at 1000 rpm at 80 °C for 50 min. Then, the stirring speed was enhanced to 1400 rpm and a mixture of 100% *E*-BTSE (704.34 mg, 2.00 mmol) with **Pn** (0.2 mmol) were added rapidly under stirring. The condensation process was conducted for two hours at 80 °C. Afterwards, the suspension was cooled to room temperature while stirring and the NPs were collected by centrifugation during 45 min at 13,500 rpm. The samples were then extracted three times with a solution of NH_4_NO_3_ (6 g/L in 96% EtOH), and washed three times with 96% ethanol, Mili-Q water, 96% ethanol, respectively. Extraction and the following steps were identical to those described for **E PMONPs**. The **E-Pn 90/10 PMONPs** were obtained as a white solid. **E-P1 90/10 PMONPs**: ^13^C-CP-MAS NMR (75 MHz) *δ*: 155.6, 146.1, 123.8, 81.1, 57.9, 52.5, 28.2, 24.9, 17.4, 10.4. IR *ν* (ATR) (cm^−1^): 3337.7, 1671.3, 1416.6, 1187.7, 1036.9, 927.3. BET: S_BET_ = 683 m^2^·g^−1^, V_pore_ = 0.48 cm^3^·g^−1^, ∅_pore_ = 2.8 nm. TGA (air, 5 °C/min, 20–1000 °C) residual mass 78%. Zeta Potential: ζ = −28.7 mV, pH = 6.99. DLS: 447 nm. **E-P2 90/10 PMONPs**: ^13^C-CP-MAS NMR (75 MHz) *δ*: 155.4, 146.0, 125.2, 81.0, 58.1, 52.8, 48.2, 28.5, 20.6, 10.8. **IR *ν* (ATR) (cm^−1^)**: 3322.9, 1673.0, 1412.0, 1187.7, 1036.2, 923.7, 187.9. **BET:** S_BET_ = 509 m^2^·g^−1^, V_pore_ = 0.34 cm^3^·g^−1^, ∅_pore_ = 2.7 nm. TGA (air, 5 °C/min, 20–1000 °C) residual mass 75%. Zeta Potential: ζ = 32.6 mV, pH = 5.92. DLS: 800 nm. **E-P3 90/10 PMONPs**: ^13^C-CP-MAS NMR (75 MHz) *δ*: 156.1, 146.1, 124.7, 81.0, 58.0, 48.0, 28.4, 18.5, 10.6. IR *ν* (ATR) (cm^−1^): 3361.9, 1655.6, 1370.1, 1188.0, 1033.4, 924.1, 788.8. BET: S_BET_ = 505 m^2^·g^−1^, V_pore_ = 0.66 cm^3^·g^−1^, ∅_pore_ = 2.8 nm. TGA (air, 5 °C/min, 20–1000 °C) residual mass 72%. Zeta Potential: ζ = 36.0 mV, pH = 5.44. DLS: 479 nm.

### 3.8. Preparation of **E-Pn 75/25 PMONPs**

In an open 250 mL round-bottom flask without reflux condenser a solution of CTAB (250 mg, 0.686 mmol) was placed in Mili-Q water (120 mL) and then 875 µL of 2 M NaOH was added (1.75 mmol of NaOH). The mixture was stirred at 1000 rpm at 80 °C for 50 min. Then, the stirring speed was enhanced to 1400 rpm and a mixture of 100% *E*-BTSE (581.09 mg, 1.65 mmol) with **Pn** (0.55 mmol) were added rapidly under stirring. The condensation process was conducted for two hours at 80 °C. Afterwards, the suspension was cooled to room temperature while stirring and the NPs were collected by centrifugation during 45 min at 13,500 rpm. The samples were then extracted three times with a solution of NH_4_NO_3_ (6 g/L in 96% EtOH), and washed three times with 96% ethanol, Mili-Q water, 96% ethanol, respectively. Extraction and the following steps were identical to those described for **E PMONPs**. The **E-Pn 75/25 PMONPs** were obtained as a white solid. **E-P1 75/25 PMONPs**: ^13^C-CP-MAS NMR (75 MHz) *δ*: 156.1, 145.5, 123.6, 81.9, 57.7, 52.8, 28.3, 24.8, 17.9, 10.4. IR *ν* (ATR) (cm^−1^): 3302.3, 1670.8, 1414.8, 1186.9, 1012.0, 926.6, 784.0. TGA (air, 5 °C/min, 20–1000 °C) residual mass 67%. Zeta Potential: ζ = −22.9 mV, pH = 7.34. DLS: 507 nm. **E-P2 75/25 PMONPs**: ^13^C-CP-MAS NMR (75 MHz) *δ*: 155.7, 145.5, 125.3, 80.2, 57.8, 28.4, 21.7, 11.1. IR *ν* (ATR) (cm^−1^): 3331.9, 1672.4, 1412.3, 1186.7, 1011.4, 923.1, 785.6. TGA (air, 5 °C/min, 20–1000 °C) residual mass 58%. Zeta Potential: ζ = 19.9 mV, pH = 6.25. DLS: 513 nm. **E-P3 75/25 PMONPs**: ^13^C-CP-MAS NMR (75 MHz) *δ*: 1561.2, 146.1, 124.8, 81.7, 58.2, 48.8, 28.6, 19.8, 10.3. IR *ν* (ATR) (cm^−1^): 3294.2, 1676.8, 1388.2, 1012.7, 923.4, 786.5. TGA (air, 5 °C/min, 20–1000 °C) residual mass 57%. Zeta Potential: ζ = 24.2 mV, pH = 4.85. DLS: 321 nm.

### 3.9. Synthesis of **P3 PMONPs**

In an open 100 mL round-bottom flask without reflux condenser a solution of CTAB (125 mg, 0.34 mmol) was placed in Mili-Q water (60 mL) and then 437 µL of 2 M NaOH was added. The mixture was stirred at 1000 rpm at 80 °C for 50 min. Then, the stirring speed was enhanced to 1400 rpm and **P3** (0.1 mmol) in absolute ethanol (1.0 mL) was added rapidly under stirring. The condensation process was conducted for 24 h at 80 °C. Afterwards, the suspension was cooled to room temperature while stirring and the NPs were collected by centrifugation during 15 min at 20,000 rpm. The samples were then extracted twice with a solution of NH_4_NO_3_ (6 g/L in EtOH), and washed three times with ethanol, Mili-Q water, ethanol, respectively. Extraction and the following steps were identical to those described for **E PMONPs**. **P3 PMONPs** were obtained as a white solid. IR *ν* (ATR) (cm^−1^): 3395.7, 2936.4, 1682.3, 1367.4, 1103.1, 1023.6, 783.8, 688.4. BET: S_BET_ = 186 m^2^·g^−1^, V_pore_ = 0.26 cm^3^·g^−1^, ∅_pore_ = 5.64 nm. Zeta Potential: ζ = 10.9 mV, pH = 8.94. DLS: 269 nm.

### 3.10. HIFU Stimulation

Nanoparticles were resuspended in PBS at a concentration of 10 mg/mL in an eppendorf tube. The tube was submitted to an ultrasound field from a focused 1 MHz probe (Precision Acoustics, UK) driven by a signal generator (Agilent 33220a, Les Ulis, France) and RF power amplifier (ADECE, Artannes sur Indre, France). The transducer was placed in a custom-made three-axis motorized positioning system to send ultrasound accurately on the eppendorf tube at the focus of the probe. Degased water was applied between the transducer and the eppendorf tube. Peak negative pressures of 1.0, 1.4 and 1.6 MPa in continuous mode were used in this study. Ultrasound stimulation time was set to 15 s. The transducer was calibrated using an HGL-200 PVDF bullet type hydrophone (Onda, Sunnyvale, CA, USA), at the focus distance of the transducer.

## 4. Conclusions

In conclusion, we studied the preparation of nanomaterials possessing a BOC group. Molecules with BOC and two to four triethoxysilyl groups were successfully synthesized through click chemistry. Then, the molecules were reacted through the sol gel procedure with bis(triethoxysilyl)ethylene, at a proportion of 10/90 to 25/75, and the materials were analyzed with different techniques. The nanorods **E-P1**, **E-P2**, **E-P3**, at a proportion of 90/10, possessed small wormlike mesopores, while, at a proportion of 75/25, the nanomaterials were non-porous at 78 K. The BOC group was not damaged during the sol gel-procedure, as shown by FTIR and ^13^C CPMAS solid-state NMR. TGA showed the stability of BOC group inside the materials, as no decomposition occurred before 120 °C. Alternatively, **P3 PMONPs** were synthesized from molecule **P3** without any other source of silicate. The obtained nanoparticles were of 100 nm diameter, and were mesoporous with a porosity of 5.6 nm. The BOC was very stable as we needed to adjust the pH to 1 to cleave the BOC at 100 °C. We were not able to observe bubbles at pH 5.5 under our conditions of HIFU, but the concept is nevertheless promising for future contrast agents for HIFU. Indeed, the electrophilicity of BOC has been shown to strongly influence the reaction rate and reactivity of the cleavage. [37] Therefore, the synthesis of precursor molecules possessing more electrophilic BOC groups can be considered. Another possibility is to introduce carboxylic acid functions in the PMONPs using triethoxysilanes possessing this function [38] during the sol-gel procedure, in order to facilitate BOC cleavage in the materials.

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
