# Peer review of "Periodic Mesoporous Organosilica Nanoparticles with BOC Group, towards HIFU Responsive Agents"

_molecules, 2020, doi:10.3390/molecules25040974_

Round 1

Reviewer 1 Report

Manuscript chemistry wise sounds good. However, I felt it application is highly recommended to present in the paper as a contrast agent. Otherwise, we do not know the importance of these compounds.

Author Response

Manuscript chemistry wise sounds good. However, I felt it application is highly recommended to present in the paper as a contrast agent. Otherwise, we do not know the importance of these compounds.

Answer: We added a paragraph with two references in the conclusion in order to improve the cleavage of BOC under HIFU:

Indeed the electrophilicity of BOC has been shown to strongly influence the reaction rate and reactivity of the cleavage. [37] Therefore, the synthesis of precursor molecules possessing more electrophilic BOC groups can be considered. Another possibility is to introduce carboxylic acid functions in the PMONPs using triethoxysilanes  possessing this function,[38] during the sol-gel procedure, in order to facilitate BOC cleavage in the materials.

Reviewer 2 Report

The manuscript by J.O. Durand et al., reports a new application of PMONPs synthesized from BOC-functionalized molecules.The manuscript can be accepted with minor revisions.

The PMONPs characterization is well detailed and described but the applicability in nanomedicine of these NPs in combination with HIFU is lacking because of the very low pH and high temperature at which the BOC group is cleaved. 

If possible, the authors should discuss improvements and future prospectives to validate their concept as promising.

Author Response

The manuscript by J.O. Durand et al., reports a new application of PMONPs synthesized from BOC-functionalized molecules.The manuscript can be accepted with minor revisions.

The PMONPs characterization is well detailed and described but the applicability in nanomedicine of these NPs in combination with HIFU is lacking because of the very low pH and high temperature at which the BOC group is cleaved. 

If possible, the authors should discuss improvements and future prospectives to validate their concept as promising.

Answer: We added a paragraph with two references in the conclusion in order to improve the cleavage of BOC under HIFU cf : reviewer 1.

Reviewer 3 Report

The article concerns novel mesoporous specimens: three bis(triethoxysilylated) tetra(triethoxysilylated) molecules. The article can be quite interesting for chemists, especially for those dealing with nanotechnology. Nevertheless, the article is hard to read in the present form.

The abstract should be rewritten more conventionally, with no division into a „chapters”. Moreover, if you use an abbreviation, it should be clarified before (HIFU).

The introduction is written as a single paragraph. The authors should divide it into at least two: the state of the art. And the scope of the article.

The nomenclature of the samples are terrific – I cannot understand, what sample is described, what sample is connected to the plot… What are the „nanorods from E”? I tried to understand, what sample is described in Fig. 3 (for example, this presenting Bragg peaks), but I’m not able to do this. I think that the average reader also can have difficulties with this.

Authors maintain that some of the samples have ordered porous structure. This is not visible under TEM microscopy – why? Pores with a diameter of 2nm are clearly visible under TEM. Explain this and replace TEM pictures with better ones, with visible porosity. The ordered hexagonal structure should be evident.

Figure 3 – the caption is too small – I cannot read the description. Enlarge this, please.

After suggested corrections, the article can be reconsidered for publication, since the content is quite valuable.

Author Response

The article concerns novel mesoporous specimens: three bis(triethoxysilylated) tetra(triethoxysilylated) molecules. The article can be quite interesting for chemists, especially for those dealing with nanotechnology. Nevertheless, the article is hard to read in the present form.

The abstract should be rewritten more conventionally, with no division into a „chapters”. Moreover, if you use an abbreviation, it should be clarified before (HIFU).

Answer: The abstract was written following the rules of MDPI with 4 subdivisions as required. HIFU was explained.

The introduction is written as a single paragraph. The authors should divide it into at least two: the state of the art. And the scope of the article.

Answer: The reviewer is right and three paragraphs have been made.

The nomenclature of the samples are terrific – I cannot understand, what sample is described, what sample is connected to the plot… What are the „nanorods from E”? I tried to understand, what sample is described in Fig. 3 (for example, this presenting Bragg peaks), but I’m not able to do this. I think that the average reader also can have difficulties with this.

Answer: We are sorry for this. The nomenclature is now explained in the introduction. 

Authors maintain that some of the samples have ordered porous structure. This is not visible under TEM microscopy – why? Pores with a diameter of 2nm are clearly visible under TEM. Explain this and replace TEM pictures with better ones, with visible porosity. The ordered hexagonal structure should be evident.

Answer: The reviewer is right and the TEM images have been done and replaced, showing the mesoporosity of the samples  E-Pn PMONPs. Note that E-Pn PMONPs showed wormlike porosity and not an ordered hexagonal structure, except for E PMONPs as displayed by small angle XRD.

Figure 3 – the caption is too small – I cannot read the description. Enlarge this, please.

Answer: The caption has been changed.

After suggested corrections, the article can be reconsidered for publication, since the content is quite valuable.

In addition we have corrected the TGA percentages.

Round 2

Reviewer 1 Report

Revision was satisfactory

Reviewer 3 Report

The manuscript in the present form is much better - (almost) all my remarks have been remarked correctly. Especially new TEM images increase the scientific value of the text. 

I recommend to accept the manuscript in the present form